# Antibacterial and Anti-Inflammatory Properties of Peptide KN-17

**DOI:** 10.3390/microorganisms10112114

**Published:** 2022-10-26

**Authors:** Qian Zhang, Shuipeng Yu, Meilin Hu, Zhiyang Liu, Pei Yu, Changyi Li, Xi Zhang

**Affiliations:** 1School and Hospital of Stomatology, Tianjin Medical University, 12 Observatory Road, Tianjin 300070, China; 2College of Electronic Information and Optical Engineering, Nankai University, 38 Tongyan Road, Tianjin 300350, China; 3Department of Prosthodontics, Affiliated Stomatology Hospital of Guangzhou Medical University, 39 Huangsha Avenue, Guangzhou 510150, China

**Keywords:** antibacterial activity, anti-inflammatory activity, antimicrobial peptide, NF-κB signaling pathway, peri-implantitis

## Abstract

Peri-implantitis, an infectious disease originating from dental biofilm that forms around dental implants, which causes the loss of both osseointegration and bone tissue. KN-17, a truncated cecropin B peptide, demonstrated efficacy against certain bacterial strains associated with peri-implantitis. This study aimed to assess the antibacterial and anti-inflammatory properties and mechanisms of KN-17. The effects of KN-17 on oral pathogenic bacteria were assessed by measuring its minimum inhibitory concentration (MIC) and minimum bactericidal concentration (MBC). Moreover, the cytotoxicity and anti-inflammatory effects of KN-17 were evaluated. KN-17 inhibited the growth of *Streptococcus gordonii* and *Fusobacterium nucleatum* during in vitro biofilm formation and possessed low toxicity to hBMSCs cells. KN-17 also caused RAW264.7 macrophages to transform from M1 to M2 by downregulating pro-inflammatory and upregulating anti-inflammatory factors. It inhibited the NF-κB signaling pathway by reducing IκBα and P65 protein phosphorylation while promoting IκBα degradation and nuclear P65 translocation. KN-17 might be an efficacious prophylaxis against peri-implant inflammation.

## 1. Introduction

Peri-implant disease occurs in response to oral biofilm formation. The incidence of this condition is increasing annually [1]. The incidence of peri-implant inflammation at 5–10 years after implantation reportedly may be as high as 20% [2]. Hence, there is a high risk of implant failure [3]. The “cuff” structure formed by the gingival (soft tissue) attachment of the implant can act as a biological barrier. There are only minor morphological differences between the soft tissue around the implant “cuff” and the natural teeth [4]. Nevertheless, the former has significantly lower resistance to plaque biofilm and periodontal pathogens than the latter. Thus, the implant “cuff” has a weaker sealing effect than natural teeth and the former is relatively more susceptible than the latter to mechanical- and toxin-induced inflammation [5]. Bacterial adhesion provokes an inflammatory reaction that acts on the implant-bone interface and results in the loss of osseointegration and bone tissue [6].

Local administration of antibacterial drugs at the implant “cuff” may effectively reduce the attachment of periodontal pathogens, promote soft tissue sealing at the gingival part of the implant [7], reduce the incidence of oral biofilm-related diseases, and achieve long-term implant stability. Nevertheless, antibiotic overuse has induced resistance in many bacterial pathogens. Therefore, it is crucial to find alternative methods of controlling biofilm-associated peri-implantitis that do not readily cause pathogen resistance. Antimicrobial peptides have received substantial research attention recently because microbial pathogens do not easily acquire resistance to them [8]. Cecropins have good antibacterial efficacy, are the most widely studied antimicrobial peptide [9], and promote osteoblast growth and differentiation [10]. Cecropins are α-helical proteins and small linear cationic peptides. Their NH_2_-terminal α-helices are amphiphilic while their -COOH terminal α-helices are hydrophobic. The amphiphilic property of cecropins directly affects their modes of action [11]. Cecropin B occurs in nature and comprises 35 amino acids. It possesses antifungal and antiviral properties [12,13,14] as well as activity against both Gram-positive and Gram-negative bacteria by binding bacterial cell membranes and creating pores in them. Cecropin B has the strongest activity against Gram-negative bacteria [15] and could, therefore, prevent and/or control peri-implant disease.

The interactions of charged amino acids significantly affect protein structure and function [16]. The mechanism by which the number of methylene groups affects protein structure stability is unclear. However, a prior study confirmed that truncated short peptides have greater antimicrobial efficacy than full-length peptides [17,18]. The properties of the amino acids themselves and their effects on the spatial peptide structure also affect the functional characteristics of the peptide. Singh et al. reported that the peptide fluorenylmethoxycarboyl-phenylalanine (Fmoc-F) formed by Fmoc has good antibacterial activity [19]. We speculate that this amino acid is implicated in the antimicrobial function of the peptides containing it.

Based on the foregoing research results, we designed a truncated cecropin B peptide KN-17 and explored the structure of KN-17 to evaluate its antibacterial performance and cytotoxicity. We also investigated the effects of KN-17 on macrophage polarization, immune response, and inflammatory factors. Future research may validate whether KN-17 can act on the gingival tissue associated with the implant, rapidly create a soft tissue sealing effect, and promote healing in the implant “cuff”.

## 2. Materials and Methods

### 2.1. Basic and Antimicrobial Properties of Peptides

#### 2.1.1. Peptide Synthesis

A truncated cecropin B peptide KN-17 was synthesized by Shanghai Deyi Biotechnology Co., Ltd. (Shanghai, China) and confirmed by mass spectrometry (MS). High-performance liquid chromatography (HPLC) indicated that the finished product had >95% purity. KN-17 powder was dissolved in sterile phosphate-buffered saline (PBS) to prepare a concentrated stock solution and various dilutions. The basic properties and structural information for KN-17 were obtained by using a peptide property calculator (http://www.pepcalc.com/, accessed on 1 February 2021). Alphafold v. 2.1 software (https://alphafold.com/, accessed on 1 February 2021) was used to predict the secondary structure of the peptide.

#### 2.1.2. Raman Spectroscopy

Five mg samples were measured with a Raman spectrometer (ESCALAB 250Xi; Thermo Fisher Scientific, Waltham, MA, USA) using 785 nm laser excitation, 0.43 eV limiting energy resolution, >5 × 10^−10^ mbar analytical chamber vacuum, and 0–5000 eV energy analysis range.

### 2.2. Antibacterial Activity Test

#### 2.2.1. Minimum Inhibitory Concentration (MIC) and Minimum Bactericidal Concentration (MBC)

The ability of KN-17 to inhibit bacterial growth was determined using a broth-based microdilution method [20,21]. *S. gordonii* (ATCC 10558) was cultured on brain heart infusion (BHI) agar plates. *F. nucleatum* (ATCC 25586) was cultured completely anaerobic on CDC anaerobic blood agar plates. Single colonies of *S. gordonii* and *F. nucleatum* were selected and cultured in BHI liquid medium for 24 h and 48 h, respectively. Various KN-17 solutions (10–2500 μg/mL serial double dilution) were mixed with BHI (1 × 10^6^ CFU/mL) in a 1:1 ratio and inoculated in 96-well plates. The total volume of each final mixture was 200 μL. The control group comprises various KN-17 dilutions mixed with sterile BHI medium. Optical density 600 (OD_600_) was measured by a microplate reader to confirm the bacterial growth of the mixture after 24 h anaerobic incubation of *S. gordonii*/48 h incubation of *F. nucleatum*. MIC was expressed as the lowest concentration of peptide that stopped all bacterial growth. Ten μL liquid was transferred from the wells to blood agar plates to determine MBC and incubated at 37 °C for 48 h. MBC was the concentration of peptide when >99.9% of all bacteria were inhibited.

#### 2.2.2. Biofilm Susceptibility

In our study, the microdilution method was used to evaluate the effect of peptide on biofilm formation. The biofilm of bacteria (1 × 10^6^ CFU/mL) was incubated for 24 h. And then KN-17 (80 μg/mL, *S. gordonii*)/(90 μg/mL, *F. nucleatum*) and various bacteria were incubated in 96-well plates for 24 h (*S. gordonii*)/48 h (*F. nucleatum*). Biofilm was fixed with 95% (*v*/*v*) methanol, stained with 0.5% (*w*/*v*) crystal violet, and dissolved in ethanol [21]. Then, its OD_600_ was measured.

#### 2.2.3. Confocal Laser Scanning Microscopy (CLSM)

Each 1 × 10^6^ CFU/mL bacterial suspension was mixed with 900 µL BHI and cultured in 24-well microtiter plates for 24 h (*S. gordonii*)/48 h (*F. nucleatum*) to form biofilms [21,22]. The culture media and unattached bacterial cells were removed, KN-17 (MIC) was added to saliva or serum, and the samples were cultured at 37 °C for 24 h. The adherent bacteria were stained with acridine orange/ethidium bromide (AO/EB) solution (Solarbio, Beijing, China) and enumerated under a CLSM (Leica SP8; Leica Biosystems AG, Wetzlar, Germany). For each biofilm, three representative areas per lens were scanned in ≥3 separate experiments. Images were captured with Metamorph software (Universal Imaging, West Chester, PA, USA).

#### 2.2.4. Scanning Electron Microscopy (SEM)

SEM was used to examine the morphology of *S. gordonii* and *F. nucleatum* in the presence of KN-17. The bacteria (1 × 10^6^ CFU/mL) were incubated with KN-17 in liquid medium for 24 h and the co-culture was centrifuged to obtain the bacterial precipitate. Then, 1 mL of 2.5% (*v*/*v*) glutaraldehyde was added and the cells were fixed at 4 °C for 2 h and centrifuged. The bacterial pellets were dehydrated by alcohol gradient (30%, 50%, 70%, 80%, 90%, 95%, and 100% for 15 min/concentration), lyophilized, sprayed with gold, and observed under SEM.

### 2.3. Human Bone Marrow Stromal Cells (hBMSCs) Cultured with KN-17 Peptide

#### 2.3.1. Cell Culture

The hBMSCs were purchased from Cyagen Bioscience (Guangzhou, China) and cultured in Dulbecco’s Modified Eagle’s Medium (DMEM) containing 10% (*v*/*v*) FBS, 100 U/mL streptomycin, and 100 U/mL penicillin.

#### 2.3.2. Cell Proliferation Assay

The hBMSCs were seeded in 24-well plates at a density of 500/well. A certain concentration of KN-17 was added to the medium of the experimental treatment while PBS of equivalent volume was added to the control treatment. The medium was changed every 3 d. Cell proliferation was observed under an electron microscope every 12 h. When significant changes were detected, the cells were stained with Giemsa dye and photographed with a camera [23].

#### 2.3.3. Cell Migration Assay

The hBMSCs cells were inoculated in 12-well plates at 2 × 10^5^/well density and cultured in ordinary medium until they adhered to the wall and covered the entire bottoms of the well plates. A ruler was used to guide the drawing of a vertical line at the center of each well with a sterile, high-temperature, 1 mL gun head. The scratches were rinsed thrice with PBS to remove dead cells and observed under a microscope. The fixed area of each well was selected and marked on the back of the well plate with a marker pen. PBS and KN-17 were added to the well plates. Microphotographs were taken at 0 h, 6 h, 12 h, 24 h, and 48 h and the cell migration rates were determined [24].

### 2.4. RAW264.7 Macrophages Cultured with KN-17

#### 2.4.1. Cell Culture

RAW264.7 cells (American Type Culture Collection (ATCC), Manassas, VA, USA) were cultured in DMEM (Thermo Fisher Scientific) supplemented with 10% (*v*/*v*) fetal bovine serum (FBS) and 1% (*v*/*v*) penicillin/streptomycin (Thermo Fisher Scientific) and set in a 5% O_2_ incubator (Thermo Fisher Scientific) at 37 °C. The cells were then inoculated in a 24-well culture plate at 3 × 10^4^/well. After 24 h, lipopolysaccharide (LPS, 1 µg/mL; Sigma-Aldrich Corp., St. Louis, MO, USA) and/or peptide (64 µg/mL KN-17) were added. PBS was used for the control group and the initial time point was defined as day 0 [23]. The experiment was divided into the BLANK (control), LPS (inflammation model), and LPS + KN-17 groups.

#### 2.4.2. Cell Proliferation Assay

RAW264.7 cells were seeded in 96-well plates and incubated for 3 d. Assuming that the final cell culture medium volumes were equal in all wells, the final peptide concentrations were taken to be 0, 2, 4, 8, 16, 32, 64, 128, 256, and 512 μg/mL. Cell proliferation was measured with Cell Counting Kit-8 (CCK-8; NCM Biotechnology Co., Zhejiang, China). CCK-8 reagent (10% (*v*/*v*)) was added to the culture medium. The cells were incubated at 37 °C for 90 min and OD_450_ was measured in a microplate reader (Cytation 5; Bio-Tek Instruments, Winooski, VT, USA) to determine the peptide concentration for use in the subsequent experiments [23].

#### 2.4.3. Real-Time Polymerase Chain Reaction (RT-PCR)

To evaluate the effects of KN-17 on macrophage polarization under inflammatory and non-inflammatory conditions, RNA was extracted with TRIzol Reagent (Thermo Fisher Scientific) on days 1 and 3. The target genes were *inducible nitric oxide synthase* (*INOS*), *tumor necrosis factor-alpha* (*TNF-α*), *CD86*, *interleukin 1alpha* (*IL-1α*), *arginase-1* (*Arg-1*), *transforming growth factor-beta* (*TGF-β*)*,* and *CD206*. The internal reference gene *glyceraldehyde-3-phosphate dehydrogenase* (*GAPDH*) was the internal control. The results were expressed by the 2^−ΔΔCt^ method [23,25]. The primer sequences of the differentiation markers are shown in Table 1 [23] (http://www.origene.com/, accessed on 15 September 2021).

#### 2.4.4. Microscopic Cell Polarization Morphology

The effects of KN-17 on macrophage polarization under inflammatory and non-inflammatory conditions were evaluated by observing the cell morphology. On days 1 and 3, RAW264.7 cells in a 24-well plate were placed under an electron microscope, observed at 10× and 20× magnification, and photographed [26].

#### 2.4.5. Cell Morphology and P65 Immunofluorescence Staining

RAW267.4 cells were incubated on 24 mm × 24 mm coverslips in six-well plates at a density of 1 × 10^5^/well to observe their morphology and P65 nucleation. The culture conditions were the same as those applied to the BLANK, LPS, and LPS + KN-17 treatments for RAW264.7 macrophages. On days 1 and 3, the cells were placed in 4% (*v*/*v*) paraformaldehyde (PFA) and their membranes were ruptured with 0.25% (*w*/*v*) Triton X-100 for 3–5 min. Each cell group was then blocked with 1% (*v*/*v*) bovine serum albumin (BSA) for 30 min. P65 was stained with anti-P65 antibody (Abcam, Cambridge, UK) and goat anti-mouse immunoglobulin G (IgG) secondary antibody (Alexa Fluor 488; Thermo Fisher Scientific). Tetramethylrhodamine (TRITC-rhodamine; Thermo Fisher Scientific) and 4′,6-diamidino-2-phenylindole (DAPI; Thermo Fisher Scientific) were used to stain the cytoskeletons and nuclei, respectively. The stained coverslips were fixed on slides with sealant and kept in the dark. CLSM (Carl Zeiss AG, Baden Württemberg, Germany) was used to capture images of the cells under different light sources [23].

### 2.5. Statistical Analysis

All statistical analyses were performed in GraphPad Prism v. 8.4.2 (GraphPad Software, La Jolla, CA, USA) and recorded as the means ± standard deviation (SD). An independent-sample *t*-test and one-way analysis of variance (ANOVA) were applied to identify statistically significant differences among treatment means. The pre-experimental design was used to determine all inclusion and exclusion criteria. *p* < 0.05 was considered statistically significant.

## 3. Results

### 3.1. Peptide Properties

The physicochemical properties of KN-17 were predicted with a peptide calculator (Table 2). KN-17 comprises 17 amino acids, possesses good water solubility, a net positive charge (+6), and molecular weight (MW) = 2174.70 Da in aqueous media. Figure 1A,B indicates that the amino acid residues of KN-17 are alternately arranged in hydrophilic and hydrophobic segments and 35% of all residues are hydrophobic. Alphafold was used to predict the secondary structure of the new peptide chain and help optimize the design. It revealed that KN-17 had good helical structure and stability (Figure 1C).

### 3.2. Raman Spectroscopy

We performed Raman spectroscopy to visualize the secondary structure of KN-17. Its Raman spectrum exhibited a wave number range of 200–3000 cm^−1^ (Figure 2). The peaks at 1000 cm^−1^, 1232 cm^−1^, 1430 cm^−1^, 1663 cm^−1^, and 2923 cm^−1^ represent a skeleton β-helix [27], a β-strand [28], a CH_2_CH_3_ skeleton structure [29], an irregular curl [30], and CH_2_ and CH_3_ lipid groups [31], respectively. Based on the foregoing information, we speculated that KN-17 has a certain antibacterial potential [32] and merits further investigation.

### 3.3. Antibacterial Test

#### 3.3.1. MIC and MBC Results of Different Strains

Microdilution method against bacteria biofilm formation evaluated the antibacterial activity of KN-17 in vitro. The MIC and MBC of KN-17 against *S. gordonii* and *F. nucleatum* are listed in Table 3. The results show that KN-17 has good antibacterial and bactericidal activity, which displayed relatively significant antibacterial efficacy against *S. gordonii*. with lower MIC and MBC (80 μg/mL and 200 μg/mL, respectively).

#### 3.3.2. Biofilm Inhibition

The OD of the bacterial biofilm treated with KN-17 was lower than that of the untreated control. Biofilm resistance significantly differed between treatments (*p* < 0.05) (Figure 3A). As shown in Figure 3B, in CLSM, the green and red spots represent the living bacteria and dead, respectively. In the BLANK group, the biofilm of bacteria was dense and mainly green. However, in the KN-17 group, the biofilm was less dense in the image.

There were fewer live and more dead bacteria in the KN-17 treated biofilm than that in the BLANK group.

#### 3.3.3. Scanning Electron Microscopy (SEM)

SEM was used to observe the effect of KN-17 on different bacteria and illustrate the destructive effect of the peptide on the target cell ultrastructure (Figure 4). As KN-17 ruptured the bacterial cell membrane, it had antibacterial efficacy.

### 3.4. Peptide Biocompatibility

KN-17 biocompatibility was validated on hBMSCs cells (Figure 5). Based on the CCK-8 OD, 256 μg/mL KN-17 was not cytotoxic while 64 μg/mL KN-17 promoted cell proliferation. The foregoing results suggest that the peptide has good biocompatibility within a specific concentration range. Unless otherwise specified, the KN-17 concentration used in the present study was 64 μg/mL.

### 3.5. Wound Healing Assay

Cell migration occurred in all treatment groups after 12 h, 24 h, and 48 h cultures (Figure 6). KN-17 promoted cell migration relative to the BLANK. Cell migration was substantially promoted after 6 h culture. We found a statistically significant difference in the scratch area after 48 h culture.

### 3.6. Effect of Peptides on Pro-inflammatory and Anti-Inflammatory Gene Expression in RAW 264.7

#### 3.6.1. Non-Inflammatory Conditions

Without LPS inducing RAW 264.7 cells to polarization toward M1, Figure 7A–D depicts that in the RAW264.7 macrophages under non-inflammatory conditions. On day 1, the FC value of pro-inflammatory genes *iNOS* was lower than 0.5, which means that was significantly downregulated relative to the BLANK group. Although the lower FC values of *TNF-α*, *CD86*, and *IL1α* were less than 0.5, the *p*-value is statistically significant (** *p* < 0.01), which indicated a trend in down expression. On day 3, *CD86* and *IL1α* remained a downregulated expression (FC < 0.5). Although *iNOS* showed a trend in up expression, it was also significantly downregulated relative to the BLANK group (FC < 0.5). *TNF-α* also showed an up expression trend, and the lower FC value was less than 0.5, but there was still statistical significance between the BLANK group (*** *p* < 0.001), which indicated a trend in down expression.

Figure 7E–G shows that under non-inflammatory conditions, KN-17 significantly upregulated the anti-inflammatory genes *A**rg**1*, *TGF-**β*, and *CD206* relative to the control on days 1 and 3 (FC > 1.5). The expression levels of these genes were significantly higher on day 3 than day 1, and the difference was statistically significant (*** *p* < 0.001, and **** *p* < 0.0001).

#### 3.6.2. Inflammatory Conditions

Figure 8A–D shows the expression levels of the pro-inflammatory genes under inflammatory conditions. On days 1 and 3, compared with the BLANK group, the pro-inflammatory genes *iNOS*, *TNF-α*, *CD86*, and *IL1α* were significantly upregulated (FC > 1.5). In response to the KN-17 treatment, the pro-inflammatory genes were significantly downregulated to varying degrees relative to the LPS group (FC < 0.5) on day 1, and all the difference was statistically significant in each group (**** *p* < 0.0001). On day 3, *IL1α* was downregulated relative to the LPS group (FC < 0.5); the FC value of *iNOS*, *TNF-α*, *CD86* were more than 0.5, but the *p*-value is statistically significant (** *p* < 0.01, *** *p* < 0.001), which indicated a trend in down expression.

In the anti-inflammatory genes part, as shown in Figure 8E–G. On day 1, anti-inflammatory genes *Arg1* and *TGF-β* were downregulated after LPS addition (FC < 0.5). Although the FC value of *CD206* was more than 0.5, the *p*-value is statistically significant (**** *p* < 0.0001), which indicated a trend in down expression. In response to the KN-17 treatment, the down trend of these anti-inflammatory genes was significantly inhibited. Relative to the LPS group, *Arg1* was upregulated (F > 1.5); the FC values of *TGF-β* and *CD206* were less than 1.5, but the *p*-value is statistically significant (** *p* < 0.01, *** *p* < 0.001), which indicated a trend in up expression. On day 3, compared between the LPS and BLANK groups, although the FC values of *Arg1* and *CD206* were more than 0.5, it still had a down trend, because the *p*-value is statistically significant (*** *p* < 0.001, **** *p* < 0.0001); *TGF-β* was shown as significantly downregulated (FC < 0.5). Compared with the LPS group, however, the KN-17 treatment significantly inhibited the down trend of these anti-inflammatory genes. The ability of KN-17 to reverse the anti-inflammatory gene down trend was particularly evident on day 3, which significantly upregulated the anti-inflammatory genes *A**rg**1*, *TGF-**β*, and *CD206* (FC > 1.5), and all the differences were statistically significant (*** *p* < 0.001, and **** *p* < 0.0001) (Figure 8E–G).

### 3.7. Impact of KN-17 on In Vitro Regulation of RAW 264.7 Phenotype, Cell Morphology, and Cytoskeleton Actin Staining

#### 3.7.1. Cell Morphology

Changes in macrophage morphology usually indicate that the cells have altered their polarization state [33]. These modifications are visible under electron microscopy. Unstimulated (M0) macrophages are round and retain this morphology during M1 differentiation. However, when the macrophages undergo M2 differentiation, they gradually become spindle shaped [23].

Figure 9 shows that the cell morphology change trends were the same for all groups on days 1 and 3. In the BLANK group (Figure 9a,A), the macrophages were round, grew in clusters, and underwent no morphological changes. Nevertheless, Figure 9c,C reveals that under non-inflammatory conditions, some of the cells treated with KN-17 became elongated.

On day 1 after LPS was added to simulate inflammation, the cells underwent obvious morphological changes. They enlarged and numerous tentacles extended from their membranes. While most cells remained round, a few became elongated. By contrast, the relative differences in cell morphology between treatment groups were far more evident by day 3 (Figure 9b,B). Under inflammatory conditions, the cells treated with KN-17 were more elongated than those exposed to LPS on both days 1 and 3 (Figure 9d,D).

#### 3.7.2. Cytoskeleton Actin Staining

Figure 10 shows that the results of the cytoskeleton staining were consistent with those of the cell morphology examination. On day 1, the nuclei were prominent in the BLANK group (Figure 10a) and the cytoskeletons near the nuclei were round and retained their original shape. Under non-inflammatory conditions, the cytoskeletons of the cells in the KN-17 treatment group began to spread to varying degrees (Figure 10c). After LPS addition (Figure 10b), the cytoskeletons of most cells became round near the nuclei. For the LPS + KN-17 group (Figure 10d), the cytoskeletons were significantly expanded and elongated relative to those of the BLANK control.

On day 3, the cytoskeletons in the BLANK group expanded (Figure 10A) but nonetheless retained their original cell morphology. In the KN-17 group (Figure 10C), the cytoskeletons significantly diffused and the cells had become elongated compared with those on day 1. In the LPS group (Figure 10B), the cytoskeletons had undergone obvious round diffusion. Under inflammatory conditions in the presence of KN-17 (Figure 10D), both cytoskeleton and cell elongation significantly increased.

#### 3.7.3. KN-17 Regulated NF-κB-P65 Signaling during RAW264.7 Polarization

P65 immunofluorescence staining was performed on different cell groups to explore the potential KN-17 mechanism (Figure 11). LPS caused P65 to translocate from the cytoplasm to the nucleus and distribute within the latter. In the LPS + KN-17 group, P65 was localized mainly to the cytoplasm as the peptide significantly inhibited its translocation to the nucleus.

## 4. Discussion

Postoperative dental biomaterial and implant-related infections have gradually led to the occurrence of peri-implant disease [34]. Epidemiological investigations and analyses have indicated that the incidence of peri-implant lesions is in the range of 19–65% [35] and includes peri-implant mucositis and peri-implant inflammation [2]. Soft tissue sealing restores the continuity of the anatomical structure after dental implantation and protects the soft and hard tissues around the implant against bacterial infection. Clinical studies have shown that after dental implantation, the alveolar bone around the first thread is absorbed in the labial-buccal-lingual-palatal direction [36] and the epithelial tissue crawls to the implant root and forms a blind soft tissue bag that shelters colonizing and reproducing oral anaerobes [37]. Persistence of the host inflammatory response induced by plaque biofilm leads to the destruction of soft tissue sealing around the implant and continues to its root. The host inflammatory response affects the supporting bone tissue around the implant and results in peri-implant inflammation causing irreversible and progressive marginal bone loss [38].

Biofilm-forming bacteria are highly resistant to currently administered antibiotics. Moreover, multi-drug-resistant pathogens are prevalent in biofilms [39,40]. Hence, it is extremely difficult to prevent and treat dental implant-related infection. AMPs (antimicrobial peptides, AMPs) are short, usually cationic peptides that occur in humans, animals, and plants [41]. AMPs have low toxicity, high specificity, and short interaction time. Thus, they are unlikely to promote drug resistance [42]. AMPs, their fragments, and their derivatives also regulate the antimicrobial immune response [43]. Cecropins constitute a family of comprehensive antimicrobial peptide molecules. They have strong antibacterial efficacy against both Gram-positive and Gram-negative bacteria. Cecropins A, B, C, D, and P1 are distinguished by their unique amino acid sequences. Cecropin B-derived peptide significantly inhibits the pro-inflammatory factors *NO* (*nitric oxide*, *NO*) and *TNF-α* in macrophages [44].

Amino acid composition, net charge, isoelectric point, and secondary structure determine the biological activity of AMPs [45]. To attenuate the influence of long amino acid sequences on the physicochemical properties of the peptide, we constructed the short peptide KN-17 from cecropin B while retaining the amino acid characteristics of the latter, and our research results also confirm the view.

Raman spectroscopy is an inelastic light-scattering technology with high spatial and spectral resolution. It rapidly and noninvasively collects biochemical and structural information [46] and is widely used in biomedicine. Polymer specificity-enabled Raman spectroscopy can be used to study the spatial structures of peptides. It obtains information about the molecular composition and backbone structure of peptide chains. Software predictions and Raman spectrum results disclosed that KN-17 has a β-helix structure and antibacterial properties [47].

The MIC, MBC, and biofilm susceptibility test results revealed that KN-17 has certain antibacterial properties. SEM and CLSM were used to observe the bacteriostatic and destructive effects of KN-17 on bacterial cell membranes. The number of leucine and lysine residues may contribute to the antibacterial activity and cytotoxicity of KN-17. Based on the results of earlier studies, Pandit et al. retained the lysines, replaced the leucine with valines, and synthesized a new peptide with relatively superior biocompatibility. Therefore, both lysine and valine were equally effective at reducing the cytotoxicity of the peptide [48]. Here, KN-17 promoted cell proliferation at moderate concentrations and remained safe even at higher concentrations. We guess that one possible explanation is the high proportions of valine and lysine in the amino acid sequence. The biocompatibility results obtained here for KN-17 were consistent with those reported by Pandit et al.

Rapid healing at the implant “cuff” is the basis for the formation of a good gingival biological barrier. Cell migration is the key to this healing process [49]. In the present study, we conducted a scratch experiment to observe the effects of KN-17 on hBMSC migration. KN-17 strongly promoted hBMSC migration after 6 h. Hence, KN-17 could induce rapid soft tissue closure at the gingival part of the dental implant.

Macrophages are the main effector cells of the inflammatory response and may be polarized to M1 or M2 under the various signal stimuli that follow dental implantation. Timely transformation of M1 into M2 macrophages effectively attenuates inflammation and promotes tissue repair [50,51]. The latter is the first step at the end of the inflammatory phase and leads into the next stage of the repair phase [52]. Stimulated by LPS significantly upregulated the pro-inflammatory genes in the macrophages. However, the subsequent addition of KN-17 immediately downregulated the pro-inflammatory genes while downregulation of the anti-inflammatory genes was significantly inhibited. By day 3, the opposite trend was evident and the expression levels of certain anti-inflammatory genes were significantly higher in the KN-17 than in the control group.

*TNF-α* is the first pro-inflammatory factor to be secreted by immunocytes in response to LPS challenge. It plays important roles in the host inflammatory response [53] and is therefore considered a vital target for the treatment of inflammatory disease [54]. Lysine addition improves peptide cytocompatibility [48]. Rajasekaran et al. used lysine to modify and synthesize a novel peptide sequence and then studied its anti-inflammatory activity. The modified peptide more effectively inhibited *TNF-α* than its parent compound [55]. The foregoing results suggest that the high proportion of lysine residues in KN-17 enables it to inhibit pro-inflammatory factors. This direct correlation between amino acids and inflammatory factors provides a rationale for shortening and optimizing the sequence and defining the core functional group of the peptide.

In the present study, we investigated the effects of the peptide on phenotypic macrophage differentiation in the non-inflammatory state. KN-17 regulates macrophages and promotes their differentiation into M2. Comprehensive analysis of the regulatory effects of KN-17 on macrophages indicates that the capacity of KN-17 to cause macrophages to differentiate into M2 could enable it to modulate inflammation and accelerate the repair of inflammatory damage. Morphological observations of RAW264.7 cells confirmed this process.

NF-κB is a key inflammatory response mediator that regulates several innate and adaptive immune functions [56]. P65 is a member of the NF-κB family and a crucial regulator of pro-inflammatory cytokine transcription [57]. Lu et al. found that downregulation of the P65/NF-κB signaling pathway controls macrophage transformation from M1 to M2 and inhibits the inflammatory response [58]. Here, we used immunofluorescence to detect P65/NF-κB signaling and discovered that in the presence of inflammation, KN-17 significantly inhibited nuclear P65 translocation and P65 and IκBα protein phosphorylation. Hence, KN-17 inhibits the NF-κB signaling pathway as well as macrophage M1 differentiation but promotes macrophage M2 transformation. Zhou et al. reported that cecropin B activates NF-κB signaling [59]. However, KN-17 is an amino acid fragment derived from cecropin B. As the compounds have different spatial structures, they would also perform differently.

## 5. Conclusions

Peri-implantitis is the main cause of implant failure and occurs because of infection and immune responses induced by oral bacteria. Here, the short antibacterial peptide KN-17 was designed based on the structure of cecropin B. Though it has already demonstrated antimicrobial efficacy, its mechanism is unclear. KN-17 effectively inhibited bacterial growth and biofilm formation. It had good biocompatibility, promoted cell migration, and inhibited the activation of the NF-κB signaling pathway causing macrophage polarization to M2 and promoting inflammation. The microenvironment of an organism affects peptide properties and functions. For this reason, the prophylactic efficacy of KN-17 must be evaluated in animal models. The specific amino acid compositions and sequences in KN-17 affect the pro-inflammatory factor *TNF-α.* Therefore, future research should investigate the genetic mechanisms by which KN-17 suppresses *TNF-α* and its associated signaling pathways.

## Figures and Tables

**Figure 1 microorganisms-10-02114-f001:**
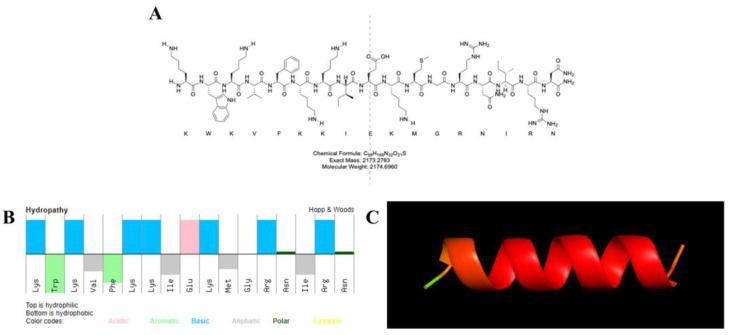
(**A**) Molecular structure formula of KN-17. (**B**) Molecular characteristics and physicochemical properties of amphiphilic peptide KN-17 (top: hydrophilic; bottom: hydrophobic). (**C**) Alphafold predicted the tertiary structures of KN-17.

**Figure 2 microorganisms-10-02114-f002:**
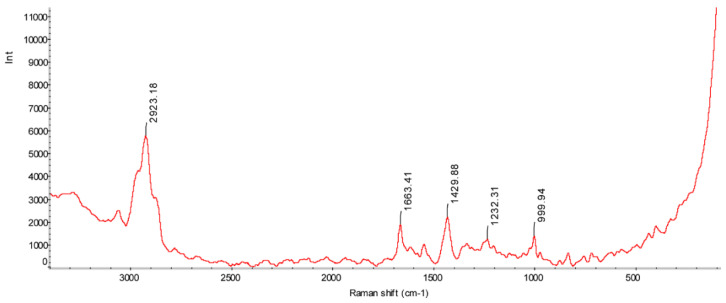
Raman spectroscopy of KN-17.

**Figure 3 microorganisms-10-02114-f003:**
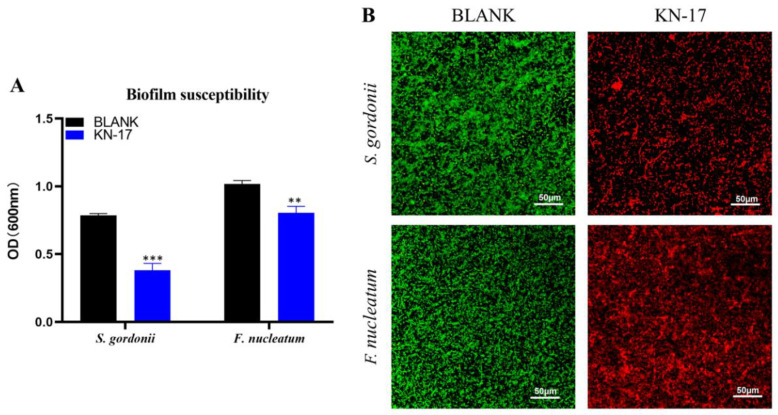
(**A**) Antibiofilm effects of KN-17 against *S. gordonii* and *F. nucleatum*, respectively. The data are presented as mean ± SD; *n* = 3. ** Indicates statistical significance between the BLANK group and KN-17 group. (** *p* < 0.01, *** *p* < 0.001, ANOVA). (**B**) The biofilms treated with KN-17 were incubated for 24 h (*S. gordonii*)/48 h (*F. nucleatum*). The biofilms were stained and imaged by CLSM. Note: Green (live bacteria), Red (dead bacteria).

**Figure 4 microorganisms-10-02114-f004:**
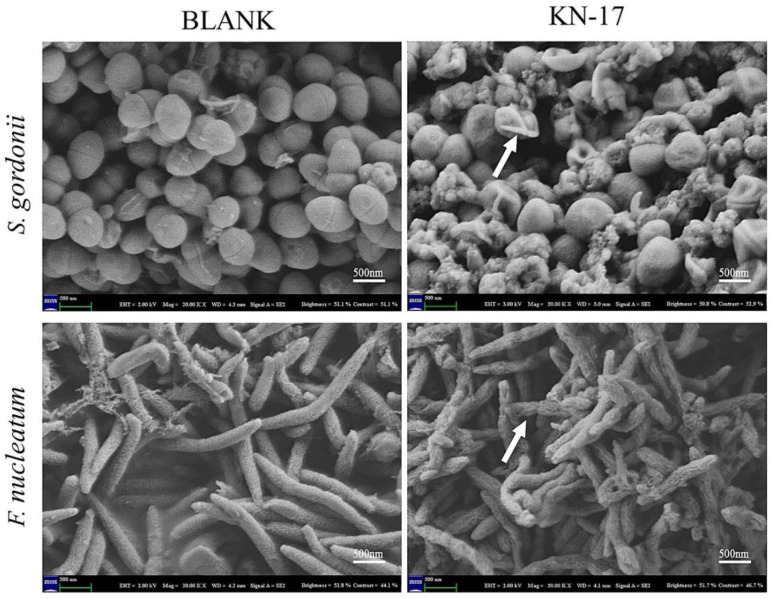
SEM images of *S. gordonii* and *F. nucleatum* biofilms treated with KN-17, respectively. The changes in bacterial morphology were observed under SEM. As indicated by the white arrows, significant cell wall distortion, corrugation, and damage were observed on the surface of *S. gordonii* and *F. nucleatum* cells after treatment with the peptide compared with the smooth surface of the control.

**Figure 5 microorganisms-10-02114-f005:**
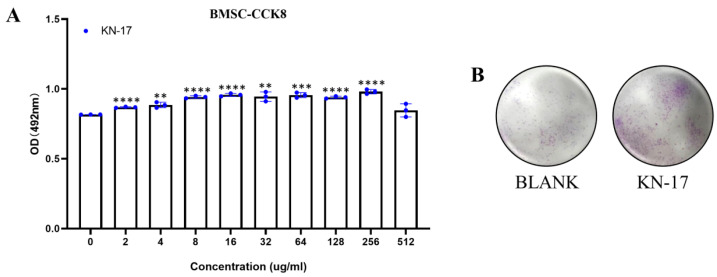
(**A**) CCK-8 results of hBMSCs cells when KN-17 was added to the culture for 3 days. Note: Data are presented as mean ± SD; *n* = 3. ** Indicates statistical significance between the BLANK group and KN-17 group. (** *p* < 0.01, *** *p* < 0.001, and **** *p* < 0.0001, ANOVA). (**B**) Cell proliferation was observed under electron microscope after adding KN-17.

**Figure 6 microorganisms-10-02114-f006:**
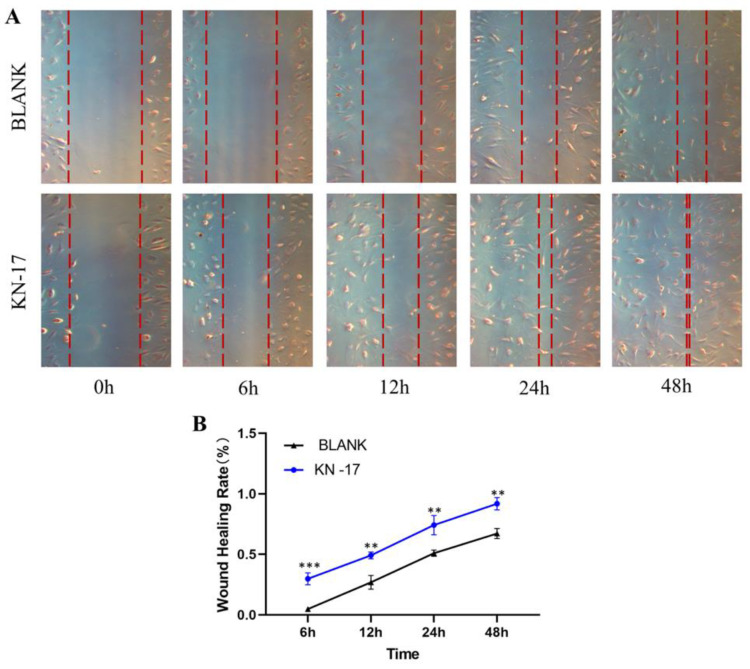
(**A**) The effects of KN-17 on the migratory capability of hBMSCs cells were measured by wound healing assay. (**B**) Wound healing rate. Note: Data are presented as mean ± SD; *n* = 3. ** Indicates statistical significance between the BLANK group and KN-17 group. (** *p* < 0.01, *** *p* < 0.001, ANOVA).

**Figure 7 microorganisms-10-02114-f007:**
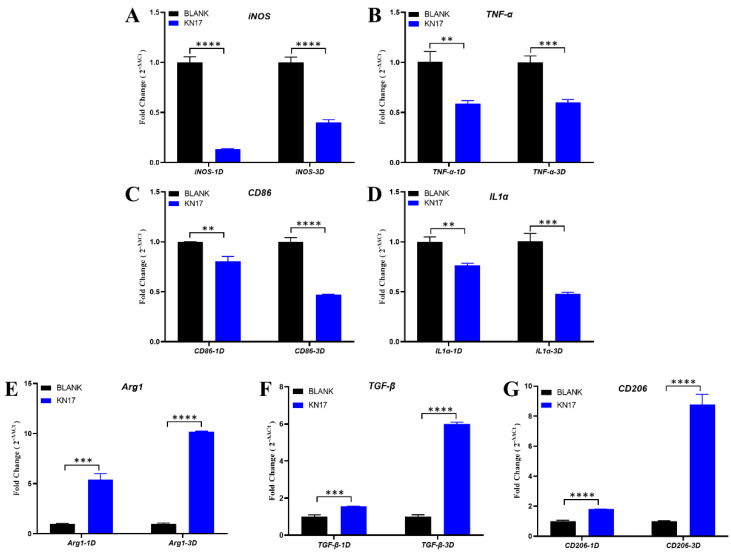
Under non-inflammatory conditions, KN-17 was added to culture RAW 264.7 cells for day 1 and day 3, respectively, and the changes in the expression of inflammation-related genes were detected by qPCR. (**A**–**D**) The levels of expression of pro-inflammatory related genes. (**E**–**G**) The levels of expression of anti-inflammatory-related genes. Note: Data are presented as mean ± SD; *n* = 3. *** Indicates statistical significance between the BLANK group and KN-17 group. (** *p* < 0.01, *** *p* < 0.001, and **** *p* < 0.0001, ANOVA).

**Figure 8 microorganisms-10-02114-f008:**
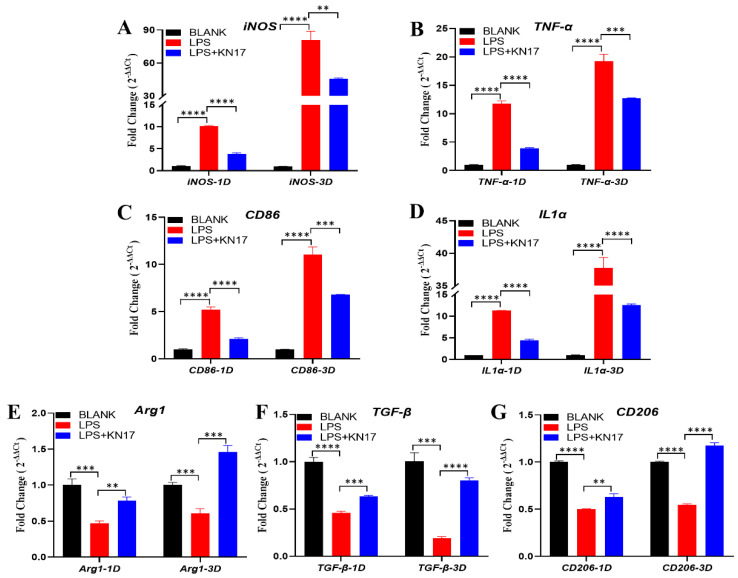
Under the conditions of LPS-induced inflammation, KN-17 was added to culture RAW 264.7 cells for day 1 and day 3, respectively, and the changes in the expressions of inflammation-related genes were detected by qPCR. (**A**–**D**) The levels of expression of pro-inflammatory related genes. (**E**–**G**) The levels of anti-inflammatory related genes. Note: Data are presented as mean ± SD; *n* = 3. ** Indicates statistical significance between the LPS group and KN-17 group. (** *p* < 0.01, *** *p* < 0.001, and **** *p* < 0.0001, ANOVA).

**Figure 9 microorganisms-10-02114-f009:**
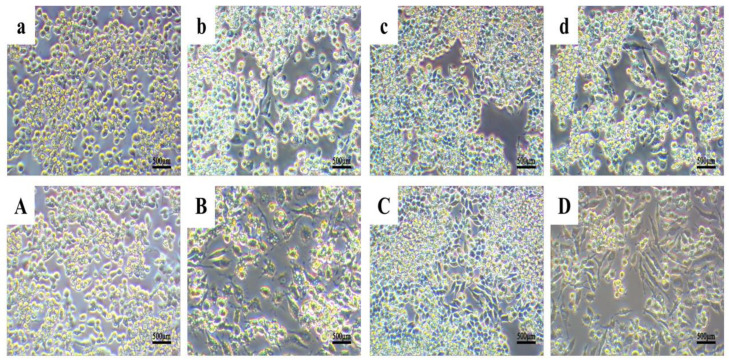
Changes in cell morphology under an electron microscope. Note: lowercase letters represent images from day 1 and uppercase letters represent images from day 3. (**a**,**A**) BLANK control group. (**b**,**B**) LPS treatment group. (**c**,**C**) KN-17 was added without LPS. (**d**,**D**) KN-17 was added at the same time.

**Figure 10 microorganisms-10-02114-f010:**
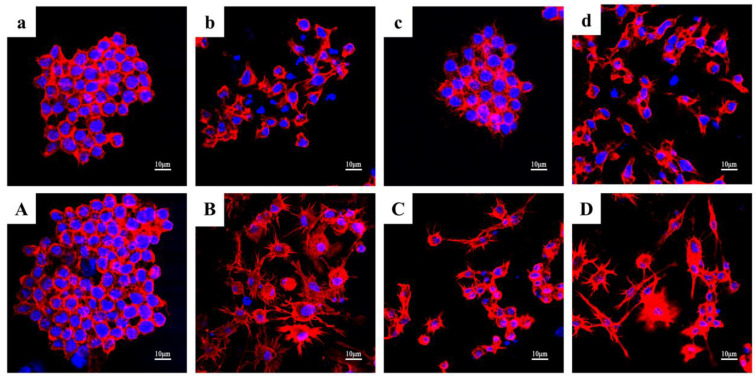
Changes in cell morphology under CLSM of RAW 264.7 cells cultured for day 1 and day 3 after different treatments. Note: lowercase letters represent images from day 1 and uppercase letters represent images from day 3. (**a**,**A**) BLANK control group. (**c**,**C**) KN-17 was added without LPS. (**b**,**B**) LPS treatment group. (**d**,**D**) KN-17 and LPS were added at the same time.

**Figure 11 microorganisms-10-02114-f011:**
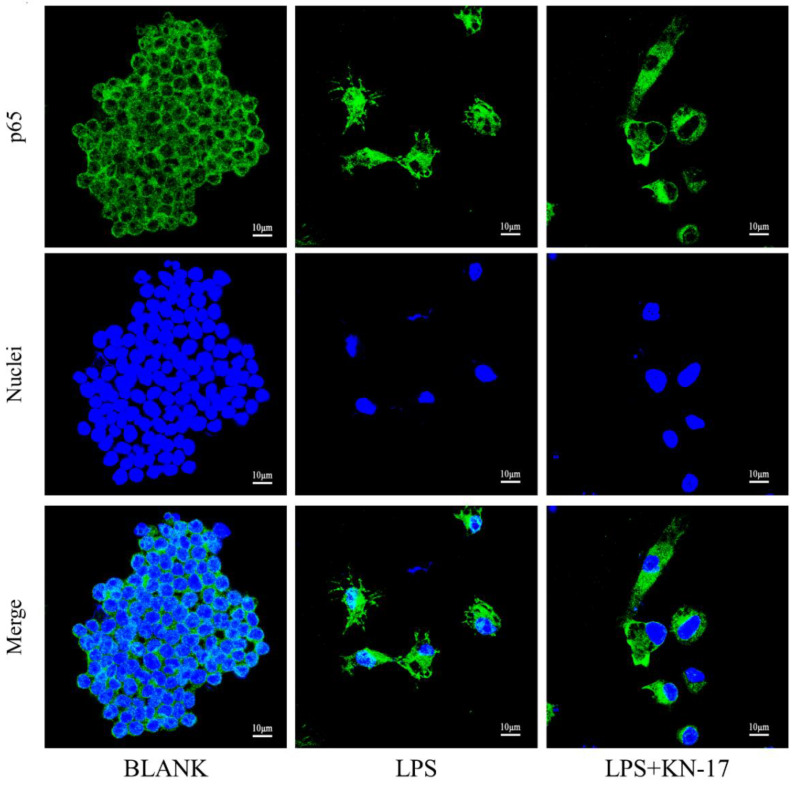
CLSM images of P65 in RAW 264.7 cells after different treatments. Green (P65); blue (nuclei).

**Table 1 microorganisms-10-02114-t001:** Primer sequences of the differentiation markers.

Gene	Forward Primer Sequence (5′-3′)	Reverse Primer Sequence (5′-3′)
*iNOS*	GAGACAGGGAAGTCTGAAGCAC	CCAGCAGTAGTTGCTCCTCTTC
*TNF-α*	GGTGCCTATGTCTCAGCCTCTT	GCCATAGAACTGATGAGAGGGAG
*CD86*	ACGTATTGGAAGGAGATTACAGCT	TCTGTCAGCGTTACTATCCCGC
*IL-1α*	ACGGCTGAGTTTCAGTGAGAC	CACTCTGGTAGGTGTAAGGTGC
*Arg-1*	CATTGGCTTGCGAGACGTAGAC	GCTGAAGGTCTCTTCCATCACC
*TGF-β*	TTGCTTCAGCTCCACAGAGA	TGGTTGTAGAGGGCAAGGAC
*CD206*	GTTCACCTGGAGTGATGGTTCTC	GTTCACCTGGAGTGATGGTTCTC
*GAPDH*	CATCACTGCCACCCAGAAGACTG	ATGCCAGTGAGCTTCCCGTTCAG

**Table 2 microorganisms-10-02114-t002:** Molecular characteristics of KN-17.

Peptide	MW	E coef	Charge	PI
KN-17	2174.70	5690	+6	11.63

**Table 3 microorganisms-10-02114-t003:** The respective MIC and MBC values of KN-17 against *S. gordonii* and *F. nucleatum*.

Bacteria	MIC (μg/mL)	MBC (μg/mL)
*S. gordonii*	80	200
*F. nucleatum*	90	220

## Data Availability

All data included in this study are available upon request through contact with the corresponding author.

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
