# Peer review of "Antibacterial and Anti-Inflammatory Properties of Peptide KN-17"

_microorganisms, 2022, doi:10.3390/microorganisms10112114_

Round 1

Reviewer 1 Report

This article investigated the antibacterial and anti-inflammatory properties of a novel peptide KN-17, which is designed by the authors based on cecropin B and their amino acid composition. The authors made a careful evaluation on the antibacterial performance, cytotoxicity, effects on macrophage polarization, the immune response, inflammatory factors of KN-17, and so on. The article is well-written, and I believe it may attract much interest from the readers in the related field. I would like to recommend the publication of the paper after minor revision.

1. Peptide KN-17 (KWKVFKKIEKMGRNIRN) is the key molecule of this paper. The authors should provide detailed characterizations of this molecules. For example, the 1H NMR, 13C NMR and HRMS of KN-17 should be provided.

2. The secondary structure of the KN-17 is very important to its biological activity. The authors are encouraged to determine the secondary structure of KN-17 using CD.

3. Some figures can be re-arranged, such as Figure 1, Figure 3 and Figure 6.

4. Spelling should be double-checked, such as line 287, line 302.

Reviewer 2 Report

This article demonstrated the antibacterial and anti-inflammatory properties of peptide, KN-17 which was considered to contribute to the success of implant therapy. The experimental design and the results seem to be exquisite and valid, however, the following points would be considered to improve this article.

1.      Abbreviations. When used for the first time you write complete and abbreviation between brackets.

2.      In the abstract section line18 assessed by measuring its MIC, MBC, antibacterial activity corrects to assessed by measuring its minimum inhibitory concertation (MIC) and minimum bactericidal concentration (MBC).

3.      In the abstract section, line 15 deletes amino acid sequence

4.      In the introduction section, line 55and 70 deletes amino acid sequence

5.      Rephrase the 2.2.2 biofilm susceptibility section in material and methods and necessary references have been added.

6.      Section 2.3 and 2.4 in materials and methods needs references added.

7.      Rephrase lines 240, 241,242, and 243.

8.      Title of figure 3 correct to Antibiofilm effects of KN-17 against S. gordonii and F. nucleatum,

9.      The authors must be used positive control antibiotics when determining MIC and MBC of KN-17.

                   Thank you

 Prof. Gamal M. El-Sherbiny

Botany and Microbiology Department, Faculty of Science, Al-Azhar University, Cairo 11884, Egypt 

Email: gamalelsherbiny1970@azhar.edu.eg  or    gamalelsherbiny1970@yahoo.com   

    ORCID ID: http://orcid.org/0000-0003-3968-0536

Reviewer 3 Report

The manuscript entitled “Antibacterial and anti-inflammatory properties of novel peptide KN-17”, wants to test the 17 aa fragment of Cecropin B against two bacterial pathogens that form biofilms that causes peri-implantitis disease , and also test the effect of the peptide  in the macrophages in order to find new strategies to manage this disease.

General comments

May main concerns about the manuscrit are:

 I think it would be great that in we could see the activity of the complete cecropin B I understand that it can not  be done with all the experiments, but at least those referring to antimicrobial activity. 

It is necessary to go through the manuscript because there are some parts that need to be clarified to understand clearly the experimental part. There are missing some important information (for example, they don't give the references for the primers used in qPCR, and only showed one primer (?).....  You can find more detailed in the specific comments.

In the results section, I would suggest to modified some graphs and figures to get more  information at first glance

Finally, the discussion need to be checked in order to remove those affirmation that not have any support by the results or stated by other authors.

Specific comments

Line 2 and 15 – Maybe is not appropriate to name the 17 aa residue of the cecropin B a novel or derived peptide. According the usual nomenclature for this kind of peptides it could be named KN-17 (cecropin B 1-17) or a 17 aa residue of the cecropin B named KN-17 or a truncated Cecropin B peptide.  Change "novel" along the manuscript

Line 13 and 14- maybe the sentences in this lines can be jointed?

Line 18- MIC and MBC are antibacterial activity, maybe will be better... measuring its antibacterial activity (MIC, MBC)?

Line 20- the first time you cite the name of the bacterial you have to use the entire name (complete genere and specie)

Lines 62-63- The paper referenced cited here is about antiviral not antimicrobial. There are references about antimicrobials but be careful with the affirmation: the shorter the peptide, the greater the activity. Please check other papers and rewrite the sentence also in lines 66-68

Line 79- I don’t think that there is design in this

Line 107-109 change well for wells (it seems that you only spread one well)

Line 111, 125- Were peptide concentrations the MIC?

Lines 111-113. I have concerns about the biofilm. Concentration/s of the peptide in the experiments were the MIC concentration, so, there is not growth?   After 24/48 h it is adhesion of the cells but it seems that to test biofilm, more incubation days are necessary according Tabares et al .2018. Can the authors explain why they uses 48 h? And give some references? Also some reference about the procedure

Tavares LJ, Klein MI, Panariello BHD, Dorigatti de Avila E, Pavarina AC. An in vitro model of Fusobacterium nucleatum and Porphyromonas gingivalis in single- and dual-species biofilms. J Periodontal Implant Sci. 2018 Feb 27;48(1):12-21. doi: 10.5051/jpis.2018.48.1.12. PMID: 29535887; PMCID: PMC5841263.

Line 162- and KN-17 alone?  As stated in line 308 of figure 7….(results in figure 7)  

Lines 179- add reference to delta-delta Ct method. (Livak et al. 2011)

Line 181- add references for the primer sequences. Only a primer? It has to be the forward and reverse primer. Which are the efficiencies of the calibration curves? Did the authors tested they?  Nothing shows us that the different reactions could be used for delta-delta Ct method

Figure 1, the size of letters in  aa one letter in A and those on B are small, it is difficult to read

Figure 3- Line 248 at the peptide concentration used (MIC), maybe the peptides have some effect in killing? Figure 3 B, is poorly explained

Line 275- I understand that from here the peptide concentration used was 64 g/ml.  (results in 3.5, 3.6 and 3.7) . Maybe, it could be also in the 2.4.3 , 2.4.4 and 2.4.5. of the experimental section.

Section 3.6-  Lines 293-298 Please, revise the results. Are the Y axes relative expression? O Fold Change (FC)? Because if you use the delta-delta Ct it will be FC. In this case values lower 0.5 are downregulated genes, and >1.5 upregulated. (there is different meaning for statistical significance)

Line 312 control = blank? Use the same name along the manuscript

Section 3.6.2- I understood that in this case the there were two tretments: the LPS and the LPS+KN-17(condition of LPS induced + peptide).

Overall section 3.6 When using delta-delta Ct, you are always comparing treatments in relation to the controls and control will be the 1 value. So, maybe the authors could remove this value  in figures 7 and 8 and maybe join the results of figure 7 and 8? 

Figure 9- I would like to suggest to move the position of the pictures. At the upper part a, c, b and d (or a,b,c,d), (time 1) and at the bottom A;C;B;D, (or A,B,C,D) (time 2)

Figure 10- I suggest the same as for figure 9.

Line 379-  please, change the KN-17 group=the LPS+kN-17 group

Line 388- remove that (twice)

Line 409—411. This is the discussion section. The authors have to state if their results agree or not… not only state a fact from another peptide

Lines 424-427. KN-17 retains some of the antibacterial properties of Cecropin B parent compound. There are not results from Cecropin B in the manuscript, although it would have add more information and importance to the study.

Lines 427-428. How can the authors state that? I do not find any study in the manuscript to prove it 

Round 2

Reviewer 3 Report

After reading the revised version of the manuscript, although there is some improvement, I think the authors has not addressed comments some of my first in a concise way. It has been accepted forever for the scientific community that a manuscript has to contain all the data in a very accurate way in the materials and methods section that will allowed any researcher to conduce the same experiments. There is some changes in the manuscript that has surprised me a lot (about the abbreviations). But, maybe other referees have asked to do i So, I encourage the authors to go through the manuscript very carefully. I hope the authors understand my concerns, because from a researcher point of view, a manuscript is a good way to learn, so we have to be sure that all is well explained and discussed. So I recomend to do a new revision of the paper.
